# Computed Tomography Indicators for Differentiating Stage 1 Borderline Ovarian Tumors from Stage I Malignant Epithelial Ovarian Tumors

**DOI:** 10.3390/diagnostics13030480

**Published:** 2023-01-28

**Authors:** Min Hoan Moon, Hee Sun Park, Young Jun Kim, Mi Hye Yu, Sungeun Park, Sung Il Jung

**Affiliations:** 1Department of Radiology, Seoul National University Seoul Metropolitan Government Boramae Medical Center, Seoul National University College of Medicine, 5 Gil 20, Boramae-Road, Dongjak-Gu, Seoul 07061, Republic of Korea; 2Department of Radiology, Konkuk University Medical Center, Research Institute of Medical Science, Konkuk University School of Medicine, 120-1, Neungdong-ro, Gwangjin-gu, Seoul 05030, Republic of Korea

**Keywords:** CT, ovary, tumor, borderline

## Abstract

Preoperative diagnosis of borderline ovarian tumors (BOTs) is of increasing concern. This study aimed to determine computed tomography (CT) features in differentiating stage 1 BOTs from stage I malignant epithelial ovarian tumors (MEOTs). A total of 170 ovarian masses (97 BOTs and 73 MEOTs) from 141 consecutive patients who underwent preoperative CT imaging were retrospectively analyzed. Two readers independently and retrospectively reviewed quantitative and qualitative CT features. Multivariate logistic analysis demonstrated that a larger tumor size (*p* = 0.0284 for reader 1, *p* = 0.0391 for reader 2) and a smaller solid component (*p* = 0.0007 for reader 1, *p* = 0.0003 for reader 2) were significantly associated with BOTs compared with MEOTs. In the subanalysis of cases with a solid component, smaller (*p* = 0.0092 for reader 1, *p* = 0.0014 for reader 2) and ill-defined (*p* = 0.0016 for reader 1, *p* = 0.0414 for reader 2) solid component was significantly associated with BOTs compared with MEOTs. Tumor size and the size and margin of the solid component were useful for differentiating stage 1 BOTs from stage 1 MEOTs on CT images.

## 1. Introduction

Borderline ovarian tumors (BOTs) compromise up to 15–20% of ovarian epithelial neoplasms and are characterized by epithelial budding, multilayering of the epithelium, increased mitotic activity, and nuclear atypia without destructive stromal invasion [1,2,3,4]. The vast majority of BOTs are limited to the ovary at presentation with 75% being diagnosed at the International Federation of Gynecology and Obstetrics (FIGO) stage 1 [5]. BOTs also have an excellent prognosis with a 10-year survival of 97% for all stages and mostly occur in women of reproductive age. Therefore, conservative treatment such as cystectomy or unilateral salpingo-oophorectomy can be applied to young patients with high fertility. 

On the contrary, the primary treatment for malignant epithelial ovarian tumors (MEOTs) includes total abdominal hysterectomy, bilateral salpingo-oophorectomy with peritoneal washing, omentectomy, and lymph node sampling. In particular, MEOTs that are diagnosed at an early stage can metastasize not only to the contralateral ovary and peritoneal cavity but also to the retroperitoneal lymph nodes; therefore, radical treatment with comprehensive staging can be required [5,6].

Contrast-enhanced computed tomography (CT) has been the standard imaging modality for preoperative staging and detection of ovarian cancer. Recently, the good spatial resolution of CT allows for ovarian mass characterization in addition to defining the extent of disease and tumor burden.

Although several studies have investigated imaging clues for the diagnosis of BOTs, most studies have compared BOTs with late-staged MEOTs that had spread to the lymph nodes, nearby pelvic organs, or peritoneum [7,8,9,10]. Thus, established knowledge regarding the CT features of ovarian masses that differentiate stage 1 BOTs from stage 1 MEOTs is limited.

Therefore, this study aimed to evaluate CT findings that differentiate stage 1 BOTs from stage 1 MEOTs.

## 2. Materials and Methods

### 2.1. Study Population

This retrospective study was approved by the institutional review board of our medical center. Medical records at our institution were reviewed to identify patients fulfilling the following inclusion criteria: (a) surgically and pathologically confirmed FIGO stage 1 BOTs or MEOTs from 1 January 2010 to 31 December 2021. The pathological diagnosis of BOTs or MEOTs was rendered according to the 2014 World Health Organization (WHO) guidelines from WHO classification of tumors of female reproductive organs; (b) underwent preoperative contrast-enhanced CT images of the abdomen and pelvis 2 weeks before surgery. Among the 145 patients who satisfied these criteria, four patients were excluded due to a lack of available enhanced CT images. All patients underwent transabdominal or transvaginal ultrasound (US) as the first-line imaging for the following symptoms or signs: abdominal distension, asymptomatic palpable pelvic mass, lower abdominal pain, abnormal uterine bleeding, and routine gynecologic US screening. The indications for CT after the US included an undetermined ovarian mass, limited US study due to poor sonic window, or exclusion of bowel disease. 

### 2.2. CT Imaging 

CT scans were obtained on a multidetector CT (Volume Zoom; Siemens, Erlangen, Germany; LightSpeed VCT XT; GE Healthcare, Milwaukee, WI, USA; or LightSpeed Pro 16; GE Healthcare, WI, USA). The Siemens scanner was set to the following parameters: detector collimation, 4 × 2.5 mm; helical pitch, 0.8; section thickness/interval, 3 mm/1.5 mm; 120 kVp/300 mA. The LightSpeed scanners were set to the following parameters: detector collimation, 64 × 0.625 mm and 16 × 1.25 mm; helical pitch, 0.984 and 0.938; section thickness/interval, 3.75 mm/3.75 mm and 3.75 mm/3.75 mm; 120 kVp/300–500 mA and 120 kVp/200–400 mA, respectively. Intravenous contrast (iopromide, Ultravist 370; Bayer Healthcare, Berlin, Germany) was injected at a rate of 3 mL/second with a total volume of 130 mL through the antecubital vein using a mechanical injector. Bolus tracking was not applied, and scanning started 80–90 s after beginning the contrast injection. No oral contrast agent was applied. Scanning regularly covered the region from the dome of the liver to the lower vagina. Coronal reformatted images were also created using the source CT data set. For reformatted images, the slice thickness and the reconstruction interval were each set to 3.0 mm. 

### 2.3. Image Analysis

Two radiologists who were blinded to all surgical and pathological data (S.I.J. and H.S.P., with 18 and 4 years of experience, respectively, in gynecologic imaging) retrospectively and independently interpreted the CT images on a picture archiving and communication system workstation (Centricity; GE Healthcare, Milwaukee, WI, USA).

For qualitative analysis, CT images were specifically evaluated for the following findings: tumor morphology, presence of a solid component that was defined as a papillary projection or a mural nodule, presence of calcification, tumor margin (well-defined or ill-defined), thick (>3 mm) septa (present or absent), and thick (>3 mm) wall (present or absent). Based on the morphology, tumors were defined as a unilocular cyst, unilocular-solid cyst, multilocular cyst, multilocular-solid cyst, and solid mass [11]. A unilocular cyst was a cyst without septa and solid component; a unilocular-solid cyst was a unilocular cyst with a measurable solid component; a multilocular cyst was a cyst with at least one septum but no measurable solid component; a multilocular-solid cyst was a multilocular cyst with a measurable solid component; a solid mass was a mass where a solid component compromises 80% or more of the mass. In cases of masses with a solid component, the margin (well-defined or ill-defined) and enhancement (mild, moderate, and strong) of the solid component were assessed in detail. Enhancement of solid components was classified into three categories as follows: mild (less than), moderate (equal), and strong (greater than) enhancement compared with that of the uterine myometrium. For quantitative analysis, the maximum diameter of the tumor and solid components were measured by both readers.

### 2.4. Statistical Analysis

Clinical and demographic data were reported using descriptive statistics. Median with interquartile range (IQR), or mean with standard deviation were used to summarize continuous variables; frequencies and percentages were used for categoric variables. Estimates were reported with 95% exact binomial confidence intervals (CI). Continuous nonparametric variables were compared with the Wilcoxon rank sum test, and categorical variables were compared with the Chi-square test or Fisher exact test. All tests were two-sided. Univariate and multivariate logistic regression analyses were performed to construct a diagnostic model to determine the weight of each CT finding in differentiating BOTs from MEOTs. Variables with *p* values ≤ 0.05 in the univariate logistic regression analysis were chosen as variables for multivariate logistic regression analysis to determine the adjusted odds ratio (OR).

Inter-reader agreement was assessed using weighted к statistics with quadratic weights for categorical variables and intraclass correlation coefficients (ICCs) for continuous variables. The data were interpreted using the following scale: slight agreement, ≤0.20; fair agreement, 0.21−0.40; moderate agreement, 0.41−0.60; substantial agreement, 0.61−0.80; and almost perfect agreement, 0.81−1.0 [12,13]. 

*p* values ≤ 0.05 were considered to indicate statistical significance. All statistical analyses were performed with a statistical software package (MedCalc Software version 14.10.2; MedCalc, Mariakerke, Belgium).

## 3. Results

The study included 170 ovarian masses from 141 patients (97 BOTs and 73 MEOTs). The median age of the patients was 52 years (range, 15–82 years). Of these, 84 (59.2%) had BOTs and 57 (40.8%) had MEOTs. Total abdominal hysterectomy with bilateral salpingo-oophorectomy and omentectomy were performed in 63 patients (19 with BOTs and 44 with MEOTs). Unilateral salpingo-oophorectomy or cystectomy was performed in 61 patients (52 with BOTs and 9 with MEOTs). The clinical and pathologic characteristics of patients are listed in Table 1.

Comparison of the CT findings of BOTs with those of MEOTs using univariate analysis identified a statistically significant difference with regard to tumor morphology, presence of solid components, tumor size, and size of the solid component. Both readers revealed that BOTs were mainly multilocular cysts (51.5% for reader 1 and 61.9% for reader 2), whereas MEOTs were commonly multilocular-solid cysts (52.1% for both readers, *p* < 0.0001). The solid component was detected more frequently in MEOTs than in BOTs (*p* < 0.0001 for both readers). The tumor size was larger in BOTs than in MEOTs (*p* < 0.0001 for both readers), and the size of the solid component was smaller in BOTs than in MEOTs (*p* < 0.0001 for both readers) (Table 2). Multivariate logistic analysis demonstrated that tumor size and size of the solid component remained significant indicators for differentiating BOTs from MEOTs. Namely, BOTs were significantly larger than MEOTs (OR 1.10, *p* = 0.0284 for reader 1 and OR 1.09, *p* = 0.0391 for reader 2), and the solid component of BOTs was significantly smaller than that of MEOTs (OR 0.58, *p* = 0.0007 for reader 1 and OR 0.43, *p* = 0.0003 for reader 2) (Table 2) (Figure 1, Figure 2, Figure 3 and Figure 4). The area under the receiver operating characteristic curve was 0.86 (95% CI, 0.80–0.90) for reader 1 and 0.79 (95% CI, 0.71–0.83) for reader 2, respectively. 

In addition, to determine the detailed characteristics of the solid component of the tumor, we performed subgroup analysis for masses with a solid component. Reader 1 classified 39 BOTs as masses with a solid component, and reader 2 classified 29 BOTs as masses with a solid component. Whereas, readers 1 and 2 classified 63 MEOTs as masses with a solid component. Univariate analysis revealed a statistically significant difference in the tumor morphology (*p* = 0.0046 for reader 1 and *p* = 0.0175 for reader 2), the size of the solid component (*p* < 0.0001 for both readers), and the margin of the solid component (*p* < 0.0001 for reader 1 and *p* = 0.0234 for reader 2) between BOTs and MEOTs (Table 3). By multivariate analysis, the size and margin of the solid component remained significant indicators for differentiating BOTs from MEOTs. Namely, the solid component was significantly smaller in BOTs than in MEOTs (OR 0.69, *p* = 0.0092 for reader 1 and OR 0.49, *p* = 0.0014 for reader 2), and the margin of the solid component was significantly ill-defined in BOTs than in MEOTs (OR 7.17, *p* = 0.0016 for reader 1 and OR 2.89, *p* = 0.0414 for reader 2) (Table 3) (Figure 1, Figure 2, Figure 3 and Figure 4). The area under the receiver operating characteristic curve was 0.83 (95% CI, 0.74–0.90) for reader 1 and 0.80 (95% CI, 0.72–0.83) for reader 2, respectively. 

Inter-reader agreement varied between substantial and almost perfect for all CT findings (к = 0.71–0.95) (Table 4).

## 4. Discussion

We demonstrated that tumor size and the size and margin of a solid component are useful for differentiating stage 1 BOTs from stage 1 MEOTs on CT images. 

Several studies have attempted to determine the value of CT imaging for the diagnosis of BOTs, and reliable data have been established yet [8,14,15,16,17,18]. In a small study on CT features of BOTs, deSouza et al. observed that the septal thickness was lesser, and the size of the solid component was smaller in BOTs than in MEOTs; however, neither feature allowed confident differentiation [9]. Grabowska-Derlatka et al. suggested that four specific CT angiographic features of tumor vascularity, including the number of vessels in the papillary projections or solid component, serpentile configuration of vessels, microaneurysm, and arteriovenous microfistulas, differentiated BOTs from MEOTs [19]. In a study using data from serous ovarian tumors alone, multivariate analysis showed that the presence of bilateral ovarian masses, peritoneal implants, and greater solid tumor volume were significantly associated with low-grade serous carcinoma than serous BOTs [15]. In a study comparing BOTs with type 1 ovarian epithelial cancers, which are low-grade malignant tumors, CT and magnetic resonance (MR) imaging revealed that the size of the solid portion and maximum wall thickness were independent indicators for differentiating between the two groups [16]. Another study comparing micropapillary and typical serous BOTs also suggested that irregular solid margins and papillary architecture/internal branching (PA/IB) are the major morphologic patterns of typical serous BOTs on CT and MR imaging, respectively [20].

In our multivariate analysis of the total ovarian mass, the solid component was less frequent and significantly smaller in BOTs than in MEOTs. These results are in accordance with those of previous studies [8,21]. However, our data also showed that the overall tumor size of BOTs was larger than that of MEOTs, which was unlike previous studies [14,15], or similar to those of previous studies [9,22]. In fact, regarding tumor size for differentiating BOTs and MEOTs, several studies have reported overlapping or conflicting data. Considering the two mechanisms for the development of ovarian neoplasm—a stepwise pathway from benign to malignant tumors, and a de novo pathway without any premalignant lesion—we suggest that the results may vary depending on the pathological composition of the study population [3,23,24].

In the subgroup analysis of masses with a solid component, we found that in addition to the size of the solid component, the margin of the solid component was an important imaging finding on CT. Namely, an ill-defined solid component was significantly more frequently detected in BOTs than in MEOTs; we assumed that this phenomenon was probably due to the loose cellular matrix or weak vascularity of the surface of the solid component in BOTs [2,25].

There are potential criticisms of our study. First, a selected bias was inevitably due to the retrospective nature of the study. Second, there remains an issue as to whether CT can be used in women of reproductive age against radiation hazards. However, this strategy can be justified because CT is the most commonly used imaging modality for the preoperative assessment of these patients. Third, our results were based on findings obtained from non-uniform CT scanners; however, this can reflect actual clinical practice.

## 5. Conclusions

In conclusion, using multivariate regression analysis, we identified that tumor size, and the size and margin of a solid component are valuable imaging indicators for differentiating stage 1 BOTs from stage 1 MEOTs on CT. These findings can help to improve the preoperative diagnosis and facilitate the selection of appropriate surgical management of BOTs.

## Figures and Tables

**Figure 1 diagnostics-13-00480-f001:**
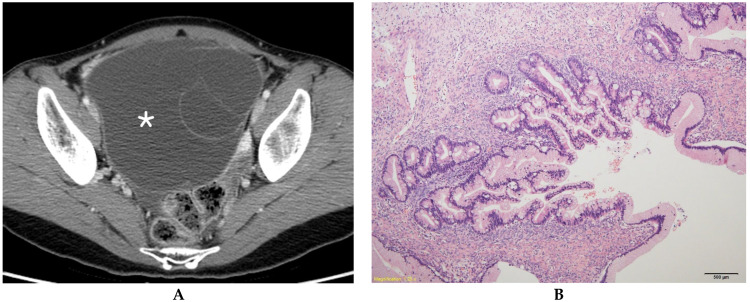
Mucinous borderline ovarian tumor in a 31-year-old woman. (**A**) Contrast-enhanced axial CT shows a multilocular cystic mass (*) without a solid component in the right ovary. (**B**) A histologic section shows aggregates of small loculi covered by a complex mucinous epithelium with a villous architecture (hematoxylin and eosin 100×).

**Figure 2 diagnostics-13-00480-f002:**
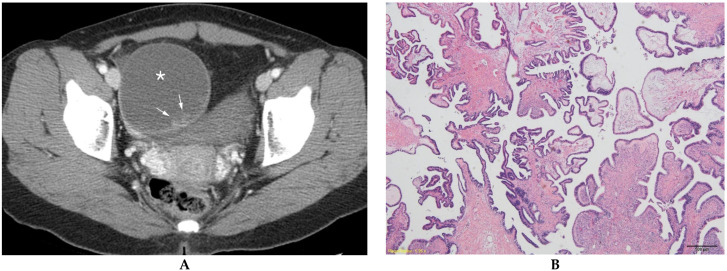
Serous borderline ovarian tumor in a 27-year-old woman. (**A**) Contrast-enhanced axial CT shows a unilocular cystic mass (*) with an ill-defined solid component (arrows) in the right ovary. (**B**) Microscopically, tumor cells present nuclear atypia and cellular proliferation without stromal invasion (hematoxylin and eosin 40×).

**Figure 3 diagnostics-13-00480-f003:**
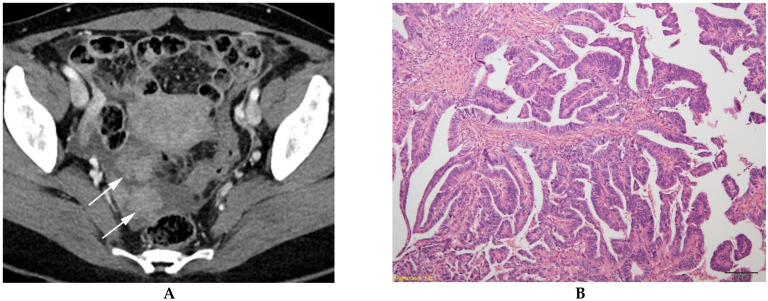
Serous ovarian carcinoma in a 44-year-old woman. (**A**) Contrast-enhanced axial CT shows a unilocular cystic mass with well-defined multiple solid components (arrows) in the right ovary. (**B**) A histologic section shows malignant glands of varying shapes and sizes infiltrating the stroma (hematoxylin and eosin 100×).

**Figure 4 diagnostics-13-00480-f004:**
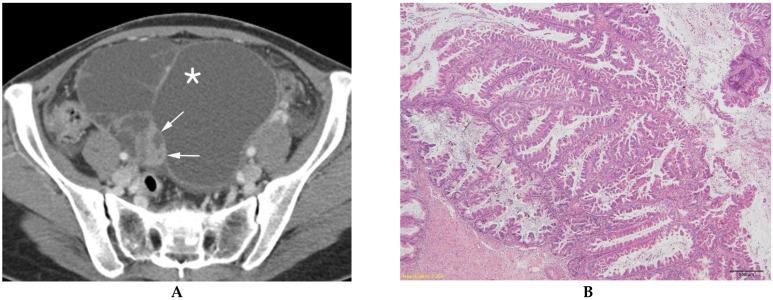
Mucinous ovarian carcinoma in a 31-year-old woman. (**A**) Contrast-enhanced axial CT shows a multilocular cystic mass (*) with a well-defined solid component (arrows) in the right ovary. (**B**) A histologic section shows malignant back-to-back glands with intracytoplasmic mucin droplets and stromal invasion (hematoxylin and eosin 40×).

**Table 1 diagnostics-13-00480-t001:** Clinical and pathologic characteristics of patients.

	BOT (n = 84)	MEOT (n = 57)	*p* Value
Median age (years)	45.5 (26.5–55.5)	55.0 (49.0–61.5)	*p* = 0.0001
Median CA-125 (U/mL)	24.1 (15.2–56.0)	75.2 (26.5–177.9)	*p* = 0.0001
Laterality			*p* = 0.09
Bilateral	13 (15.5)	16 (28.1)	
Unilateral	71 (84.5)	41 (71.9)	
Pathologic diagnosis			
Serous borderline tumor	27 (32.1)	NA	
Mucinous borderline tumor	51 (60.7)	NA	
Seromucinous borderline tumor	6 (7.1)	NA	
Serous carcinoma	NA	43 (75.4)	
Mucinous carcinoma	NA	8 (14.0)	
Clear cell carcinoma	NA	2 (3.5)	
Endometrioid carcinoma	NA	3 (5.3)	
Seromucinous carcinoma	NA	2 (3.5)	

Note—For continuous variables, data are medians with interquartile ranges in parentheses. For categoric variables, data are the number of patients with percentages in parentheses. NA = not applicable.

**Table 2 diagnostics-13-00480-t002:** Comparison of CT findings between BOTs and MEOTs.

Reader 1					Reader 2				
	BOT (n = 97)	MEOT (n = 73)	*p* Value			BOT (n = 97)	MEOT (n = 73)	*p* Value	
			Univariate Analysis	Multivariate Analysis				Univariate Analysis	Multivariate Analysis
Tumor Morphology			<0.0001	0.5376	Tumor Morphology			<0.0001	0.6780
Unilocular cyst	8/97 (8.3)	1/73 (1.4)			Unilocular cyst	8/97 (8.2)	1/73 (1.4)		
Unilocular-solid cyst	6/97 (6.2)	7/73 (9.6)			Unilocular-solid cyst	6/97 (6.2)	7/73 (9.6)		
Multilocular cyst	50/97 (51.5)	9/73 (12.3)			Multilocular cyst	60/97 (61.9)	9/73 (12.3)		
Multilocular-solid cyst	32/97 (33.0)	38/73 (52.1)			Multilocular-solid cyst	22/97 (22.7)	38/73 (52.1)		
Solid mass	1/97 (1.0)	18/73 (24.7)			Solid mass	1/97 (1.0)	18/73 (24.7)		
Calcification			1.00		Calcification			1.00	
Present	2/97 (2.1)	2/73 (2.7)			Present	2/97 (2.1)	2/73 (2.7)		
Absent (reference)	95/97 (97.9)	71/73 (97.3)			Absent (reference)	95/97 (97.9)	71/73 (97.3)		
Solid component			<0.0001	1.00	Solid component			<0.0001	1.00
Present	39/97 (40.2)	63/73 (86.3)			Present	29/97 (29.9)	63/73 (86.3)		
Absent (reference)	58/97 (59.8)	10/73 (13.7)			Absent (reference)	68/97 (70.1)	10/73 (13.7)		
Margin of tumor			0.2333		Margin of tumor			0.2537	
Ill-defined	2/97 (2.1)	4/73 (5.5)			Ill-defined	3/97 (3.1)	5/73 (6.8)		
Well-defined (reference)	95/97 (97.9)	69/73 (94.5)			Well-defined (reference)	94/97 (96.9)	68/73 (93.2)		
Wall thickening			0.5252		Wall thickening			0.3092	
Present	15/97 (15.5)	14/73 (19.2)			Present	13/97 (13.4)	14/73 (19.2)		
Absent (reference)	82/97 (84.5)	59/73 (80.8)			Absent (reference)	84/97 (86.6)	59/73 (80.8)		
Septal thickening			0.2203		Septal thickening			0.3129	
Present	20/97 (20.6)	21/73 (28.8)			Present	25/97 (25.8)	24/73 (32.9)		
Absent (reference)	77/97 (79.4)	52/73 (71.2)			Absent (reference)	72/97 (74.2)	49/73 (67.1)		
Tumor size (cm)	10.9 (6.6–14.9)	7.1 (4.2–10.4)	<0.0001	0.0284	Tumor size (cm)	10.5 (6.4–14.1)	7.3 (4.3–10.1)	<0.0001	0.0391
Size of solid component (cm)	2.1 (1.0–2.6)	3.6 (2.3–5.2)	<0.0001	0.0007	Size of solid component (cm)	1.9 (1.1–3.4)	4.0 (2.4–5.1)	<0.0001	0.0003

**Table 3 diagnostics-13-00480-t003:** Comparison of CT findings between BOTs and MEOTs for masses with a solid component.

Reader 1					Reader 2				
	BOT (n = 39)	MEOT (n = 63)	*p* Value			BOT (n = 29)	MEOT (n = 63)	*p* Value	
			Univariate Analysis	Multivariate Analysis				Univariate Analysis	Multivariate Analysis
Tumor morphology			0.0046	0.2542	Tumor morphology			0.0175	0.0979
Unilocular- solid cyst	6/39 (15.4)	7/63 (11.1)			Unilocular-solid cyst	6/29 (15.4)	7/63 (11.1)		
Multilocular-solid cyst	32/39 (82.1)	38/63 (60.3)			Multilocular-solid cyst	22/29 (75.9)	38/63 (60.3)		
Solid mass	1/39 (2.6)	18/63 (28.6)			Solid mass	1/29 (3.4)	18/63 (28.6)		
Enhancement of solid component			0.153		Enhancement of solid component			0.187	
Mild	31/39 (79.5)	43/63 (68.3)			Mild	24/29 (82.8)	41/63 (65.1)		
Moderate	6/39 (15.4)	18/63 (28.6)			Moderate	4/29 (13.8)	20/63 (31.8)		
Avid	2/39 (5.1)	2/63 (3.2)			Avid	1/29 (3.5)	2/63 (3.2)		
Margin of solid component			<0.0001	0.0016	Margin of solid component			0.0234	0.0414
Ill-defined	20/39 (51.3)	6/63 (9.5)			Ill-defined	16/29 (55.2)	16/63 (25.4)		
Well-defined (reference)	19/39 (48.7)	57/63 (90.5)			Well-defined (reference)	13/29 (44.8)	47/63 (74.6)		
Wall thickening			1.000		Wall thickening			0.370	
Present	6/39 (15.4)	11/63 (17.5)			Present	3/29 (10.3)	13/63 (20.6)		
Absent (reference)	33/39 (84.6)	52/63 (82.5)			Absent (reference)	26/29 (89.7)	50/63 (79.4)		
Septal thickening			0.650		Septal thickening			0.238	
Present	10/39 (25.6)	19/63 (30.2)			Present	7/29 (24.1)	24/63 (38.1)		
Absent (reference)	29/39 (74.4)	44/63 (69.8)			Absent (reference)	22/29 (75.9)	39/63 (61.9)		
Tumor size (cm)	8.5 (5.8–12.6)	7.1 (4.3–10.5)	0.1124		Tumor size (cm)	6.4 (5.0–10.4)	7.4 (4.4–9.9)	0.6559	
Size of solid component (cm)	2.1 (1.0–2.6)	3.7 (2.3–5.2)	<0.0001	0.0092	Size of solid component (cm)	1.9 (1.1–2.4)	4.0 (2.4–5.1)	<0.0001	0.0014

**Table 4 diagnostics-13-00480-t004:** Inter-observer agreement among readers of each CT finding for tumors.

CT Finding	Reader 1 vs. Reader 2
Tumor morphology	0.78 (0.61–0.83)
Calcification of tumor	0.93 (0.83–1.00)
Solid component or tumor	0.85 (0.71–0.93)
Tumor margin	0.95 (0.89–1.00)
Wall thickening of tumor	0.71 (0.58–0.81)
Septal thickening of tumor	0.73 (0.61–0.89)
Enhancement of solid component	0.94 (0.88–0.99)
Margin of solid component	0.79 (0.63–0.88)
Tumor size	0.85 (0.81–0.92)
Size of solid component	0.82 (0.73–0.88)

Notes—Data with 95% CIs in parentheses.

## Data Availability

Not available.

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
