# Peer review of "Computed Tomography Indicators for Differentiating Stage 1 Borderline Ovarian Tumors from Stage I Malignant Epithelial Ovarian Tumors"

_diagnostics, 2023, doi:10.3390/diagnostics13030480_

Round 1

Reviewer 1 Report

The authors conducted a case-control study to investigate computed tomography (CT) findings that differentiates stage 1 borderline ovarian tumors (BOTs) from stage I malignant epithelial ovarian tumors (MEOTs). As a result, they found that tumor size and margin of the size of the solid component were useful 24 for differentiating stage 1 BOTs from stage 1 MEOTs on CT images. The study design sounds fair. The result that a larger size of the solid component is associated with malignancy is not so attractive (easily predictable).

Abstract: OK

Introduction:

1. "Although several studies have investigated imaging clues for the diagnosis of BOTs, most studies have compared BOTs with late-staged MEOTs that had spread to the lymph nodes, nearby pelvic organs or peritoneum." - Please cite several references and explain what is already known and what is not yet more concisely.

Materials and Methods: 

Study population

2. "Ultimately, a total of 170 ovarian 65 masses (97 BOTs and 73 MEOTs) from 141 consecutive patients (age range, 15-82 years; 66 median age, 52 years) were included in this study." - This part is duplicated with a sentence in the result. Please remove.

CT imaging 

3. "CT scans were obtained on a MDCT" - "an MDCT"... no need to use abbreviation.

Results:

4. "The study included 170 ovarian masses from 141 patients (97 BOTs and 73 MEOTs). The median age of the patients was 52 years (range, 15-82 years). Of these, 84 (59.2%) had BOTs and 57 (40.8%) had MEOTs." - How many patients had both BOT and MEOT, bilateral ovarian tumors, or a unilateral collision tumor of BOT and MEOT?

5. "Unilateral salpingo-oophorectomy or cystectomy was performed in 61 patients (52 with BOTs and 9 with MEOTs). The clinical and pathologic characteristics of patients are listed in Table 1." - There seem differences between the groups on surgical procedures. On what basis was the surgical procedure determined?

6. Tables 1-3. Please reduce the number of digits for "%" to three or two digits. The number of digits for "cm" could be reduced to one decimal place because measurements down to 0.01 cm are not unrealistic.

7. Spelling error: "clssified", 

Discussion:

8. Spelling errors: reilaible, angigraphic, vasularity, BOTs form MEOTs, siginificant smaller, mecahnism, neoplasim, whitout, compostion, papulation.

9. "Considering the two mecahnism for the development of ovarian neoplasim — a stepwise pathway from benign to malignant tumors, and a de novo pathway whitout any premalignant lesion — we suggest that the results may vary depending on the pathological compostion of the study papulation" - What mechanism plays a large role based on the present results? Why BOTs became larger in size than MEOTs? Please speculate. 

Author Response

Response to Reviewer 1 comments

Point 1. "Although several studies have investigated imaging clues for the diagnosis of BOTs, most studies have compared BOTs with late-staged MEOTs that had spread to the lymph nodes, nearby pelvic organs or peritoneum." - Please cite several references and explain what is already known and what is not yet more concisely.

Response:

Previous studies for the diagnosis of BOT mainly highlighted MR features of BOT compared with MEOTs, and included the imaging features such as ascites, presence of peritoneal implants, or pattern of peritoneal dissemination, which were shown in advanced ovarian cancers. So far, we thought there was no comparison study between BOTs and MEOTs by CT in terms of early same stage (stage 1).

   And we cited following references:

  1. Bent CL, Sahdev A, Rockall AG, Singh N, Sohaib SA, Reznek RH: MRI appearances of borderline ovarian tumours. Clin Radiol 2009, 64(4):430-438.
  2. Denewar FA, Takeuchi M, Urano M, Kamishima Y, Kawai T, Takahashi N, Takeuchi M, Kobayashi S, Honda J, Shibamoto Y: Multiparametric MRI for differentiation of borderline ovarian tumors from stage I malignant epithelial ovarian tumors using multivariate logistic regression analysis. Eur J Radiol 2017, 91:116-123.
  3. deSouza NM, O'Neill R, McIndoe GA, Dina R, Soutter WP: Borderline tumors of the ovary: CT and MRI features and tumor markers in differentiation from stage I disease. AJR Am J Roentgenol 2005, 184(3):999-1003.
  4. Sahin H, Akdogan AI, Smith J, Zawaideh JP, Addley H: Serous borderline ovarian tumours: an extensive review on MR imaging features. Br J Radiol 2021, 94(1125):20210116.

Point 2. "Ultimately, a total of 170 ovarian 65 masses (97 BOTs and 73 MEOTs) from 141 consecutive patients (age range, 15-82 years; 66 median age, 52 years) were included in this study." - This part is duplicated with a sentence in the result. Please remove.

Response: Ok, We removed that.

Point 3. "CT scans were obtained on a MDCT" - "an MDCT"... no need to use abbreviation.

Response: We changed to “multidetector CT”.

Point 4. "The study included 170 ovarian masses from 141 patients (97 BOTs and 73 MEOTs). The median age of the patients was 52 years (range, 15-82 years). Of these, 84 (59.2%) had BOTs and 57 (40.8%) had MEOTs." - How many patients had both BOT and MEOT, bilateral ovarian tumors, or a unilateral collision tumor of BOT and MEOT?

Response: As mentioned in table 1, 13 patients had bilateral BOTs and 16 patients had bilateral 16 MEOTs. There was no one who had both BOT and MEOT, or a unilateral collision tumor of BOT and MEOT. 

Point 5. "Unilateral salpingo-oophorectomy or cystectomy was performed in 61 patients (52 with BOTs and 9 with MEOTs). The clinical and pathologic characteristics of patients are listed in Table 1." - There seem differences between the groups on surgical procedures. On what basis was the surgical procedure determined?

Response: Gynecologic surgeons decided an optimal surgical procedure according to preoperative imaging and clinical data, and the status of operative field. In the case of suspected patients with BOTs of reproductive age, they used to do conservative surgery if possible.

Point 6. Tables 1-3. Please reduce the number of digits for "%" to three or two digits. The number of digits for "cm" could be reduced to one decimal place because measurements down to 0.01 cm are not unrealistic.

Response: We changed to revised numbers.

Point 7. Spelling error: "clssified", 

Response: We changed to correct word.

Point 8. Spelling errors: reilaible, angigraphic, vasularity, BOTs form MEOTs, siginificant smaller, mecahnism, neoplasim, whitout, compostion, papulation.

Response: We changed to correct words.

Point 9. "Considering the two mecahnism for the development of ovarian neoplasim — a stepwise pathway from benign to malignant tumors, and a de novo pathway whitout any premalignant lesion — we suggest that the results may vary depending on the pathological compostion of the study papulation" - What mechanism plays a large role based on the present results? Why BOTs became larger in size than MEOTs? Please speculate. 

Response: Considering that BOTs was larger than MEOTs in our dataset, we suggested that majority of present MEOTs cases maybe due to de novo ovarian carcinogenesis which are rarely associated with recognizable precursor lesions and evolve rapidly. In other words, we could assume that the incidence of type 1 small high-grade carcinoma can affect the difference of size between BOTs and MEOTs

Reviewer 2 Report

This study aimed to determine computed tomography (CT) features in differentiating stage I borderline ovarian tumors from stage I malignant epithelial ovarian tumors. This study is interesting and may help differentiate boderline ovarian tumor and malignant epithelial ovarian tumor on CT scans. Although this manuscript is well-presented, I have some comments to the authors.

* The CT scans were only interpretated by two senior radiologists with 18 and 16 years of experience. To increase the generalizability, I suggest the authors to increase the radiologists with different years of experience (i.e.: two radiologists with around 5 years of experience, and two radiologists with around 10 years of experience) to interprest these images. Radiologists of other institutions may also be suggested. Further analyses and discussion are also needed.

* I suggest the authors to further analyze the prediction of these features (e.g. AUC).

Author Response

Response to Reviewer 2 comments

Point 1. The CT scans were only interpreted by two senior radiologists with 18 and 16 years of experience. To increase the generalizability, I suggest the authors to increase the radiologists with different years of experience (i.e.: two radiologists with around 5 years of experience, and two radiologists with around 10 years of experience) to interpret these images. Radiologists of other institutions may also be suggested. Further analyses and discussion are also needed.

Response: As the reviewers for CT interpretation, one reviewer (S.I.J.) is a specialist for genitourinary radiology, especially is an expert in gynecologic imaging considering his career (18 years in gynecologic imaging). Whereas, although the other reviewer (H.S.P.) has worked in an abdominal imaging for 16 years, she has worked in hepatobiliary imaging during most of her career. In fact, her working period is relatively not long about gynecologic imaging (about only 4 years). We think that both reviewers can be suitable for generalizability of image interpretation.

So, we changed to “(S.I.J. and H.S.P., with 18 and 4 years of experience, respectively, in gynecologic imaging).

Point 2. I suggest the authors to further analyze the prediction of these features (e.g. AUC).

Response: We added following results:

In page 4, “The area under the receiver operating characteristic curve was 0.86 (95% CI, 0.80-0.90) for reader 1 and 0.79 (95% CI, 0.71-0.83) for reader 2, respectively.”      

In page 8, “The area under the receiver operating characteristic curve was 0.83 (95% CI, 0.74-0.90) for reader 1 and 0.80 (95% CI, 0.72-0.83) for reader 2, respectively.”

Round 2

Reviewer 2 Report

The authors had addressed my concerns.